# The Loss-Function of *KNL1* Causes Oligospermia and Asthenospermia in Mice by Affecting the Assembly and Separation of the Spindle through Flow Cytometry and Immunofluorescence

**DOI:** 10.3390/s23052571

**Published:** 2023-02-25

**Authors:** Yuwei Zhao, Jingmin Yang, Daru Lu, Yijian Zhu, Kai Liao, Yafei Tian, Rui Yin

**Affiliations:** 1State Key Laboratory of Genetic Engineering, School of Life Sciences, Fudan University, Shanghai 200000, China; 2Shanghai WeHealth BioMedical Technology Co., Ltd., Shanghai 201318, China; 3NHC Key Laboratory of Birth Defects and Reproductive Health, Chongqing Population and Family Planning Science and Technology Research Institute, Chongqing 404100, China; 4Reproductive Medicine Research Center, Medical Research Institute, Southwest University, Chongqing 400715, China

**Keywords:** *KNL1*, immunofluorescence staining, flow cytometry, oligospermia, asthenospermia

## Abstract

*KNL1* (kinetochore scaffold 1) has attracted much attention as one of the assembly elements of the outer kinetochore, and the functions of its different domains have been gradually revealed, most of which are associated with cancers, but few links have been made between *KNL1* and male fertility. Here, we first linked *KNL1* to male reproductive health and the loss-function of *KNL1* resulted in oligospermia and asthenospermia in mice (an 86.5% decrease in total sperm number and an 82.4% increase in static sperm number, respectively) through CASA (computer-aided sperm analysis). Moreover, we introduced an ingenious method to pinpoint the abnormal stage in the spermatogenic cycle using flow cytometry combined with immunofluorescence. Results showed that 49.5% haploid sperm was reduced and 53.2% diploid sperm was increased after the function of *KNL1* was lost. Spermatocytes arrest was identified at the meiotic prophase I of spermatogenesis, which was induced by the abnormal assembly and separation of the spindle. In conclusion, we established an association between *KNL1* and male fertility, providing a guide for future genetic counseling regarding oligospermia and asthenospermia, and a powerful method for further exploring spermatogenic dysfunction by utilizing flow cytometry and immunofluorescence.

## 1. Introduction

For eukaryotic cells, the process of accurate chromosome separation is described as a delicate instrument that each slight error may cause dysregulation of cell proliferation or even disease phenotype [1,2,3,4]. The kinetochore is a macromolecular protein complex that assembles at centromeres and connects to spindle microtubules [5,6,7]. It also performs several indispensable functions in the cell-division process by linking the centromere and microtubule, providing a platform for the spindle assembly checkpoint (SAC) protein [8,9,10,11]. More than 100 kinetochore proteins from yeast to mammalian cells have been identified [8]. The study of kinetochore compositions is significant and necessary to shed light on this intriguing complex in multiple fields concerning functions, evolutions, and diseases.

*KNL1* (also known as *CASC5*, *D40*, Blinkin) [12] is one important member required for genomic stability in eukaryotes in the conserved KMN network (*KNL1* complex, *NDC80* complex, *Mis12* complex), constituting the core microtubule-binding site at the kinetochore [13,14,15]. *KNL1* regulates cells by coordinating multiple protein–protein interactions through different domains on its protein structure [16]. At the N terminal of *KNL1*, the KI motif interacts with the *Bub* complex via TPR domains [17]. At the C terminal, the predicted coiled-coil domain and RWD repeats domain binds to *Zwint1* and *Mis12*, respectively, which are required to associate with the inner kinetochore [18]. As functions of *KNL1* complex domains were subsequently revealed [19,20,21], researchers have focused on the link between them and diseases. Defects in *KNL1* function have been implicated in genome instability, leukemia, microcephaly, and colorectal cancer [22,23,24,25]. Recently, Wei et al. [26] proved that *KNL1* stabilized SAC to ensure timely anaphase entry and accurate chromosome segregation during oocyte meiotic maturation, which stimulated our curiosity about the function of this gene in reproduction. Male reproductive health research is still a hot topic, but there is no association found between *KNL1* and male reproduction at the present stage.

Flow cytometry (FCM) is well known for its traits of enabling multi-parameters and rapid quantitative analysis of single cells or other biological particles, and it is widely used in the medical field [27,28,29]. The use of FCM in sperm detection has a long history (Table 1); however, we found that the established methods were based on FCM combined with different fluorescent dyes to detect, which has high requirements for instruments and antibodies. In addition, researchers mainly focused on the abnormal morphology and biochemical indicators of sperm. Less research has focused on cell cycles. Miki et al. detected the obvious differences in the composition of haploid, diploid, and tetraploid cells in spermatogenesis and found additional cell groups in the experiment group by using flow cytometry, but they did not state the problem stages specifically [30]. In this point, we converted the dye-staining signal used for FCM into an immunofluorescence (IF) signal that could be observed by microscopy used for the detection and localization of a wide variety of antigens [31], such as *GFRα1*, *PLZF*, *C-kit*, *STRA8*, *γH2AX*, and *Sycp3*, in the case of single-dye FCM detection to further determine the specific stage of the spermatogenesis cycle.

Here, we first found that the loss of *KNL1* caused oligospermia and asthenia in mice and detected the abnormal composition of different ploidy sperm cells by FCM. The knockout of *KNL1* could lead to a decrease in the immunofluorescence signal of *GFRα1*, *PLZF*, *γH2AX*, and *Sycp3* by employing IF staining, which means the spermatocytes arrested at the meiotic prophase I of spermatogenesis. Moreover, the spindle was unable to maintain its polarity and alignment correctly in spermatocytes in GC-2 cells, which resulted in the assembly separation of the spindle and confirmed that the method of FCM plus IF was useful in detecting the periodic location of spermatogenesis problems. In a word, our study not only provides a powerful method for spermatogenic dysfunction by FCM and IF but also exposes the relationship between *KNL1* and reproductive development, which helps to determine genotypic–phenotypic associations as well as improve the diagnosis rate of male infertility.

## 2. Materials and Methods

### 2.1. Cell Lines and Cell Culture

GC-2 (CEMC, Sydney NSW, Australia) cells were purchased from the Center for Excellence in Molecular Cells Science and cultured in DMEM (ThermoFisher, Waltham, MA, USA) supplemented with 10% Fetal Bovine Serum (Gibco, Grand Island, Australia) and 1% Penicillin-Streptomycin Solution (Gibco, Grand Island, Australia). All cells used in experiments were cultured at 37 °C in a 5% CO_2_ incubator.

### 2.2. Mice Culture

Sexually mature male ICR mice (more than four weeks old) were purchased from Charles River Laboratory Animal Technology CO., Ltd. (Wilmington, MA, USA) The mice were placed in a specific pathogen-free animal room under controlled conditions (temperature, 25 ± 2 °C, humidity (50–60%), and a 12/12 h light/dark cycle. All mice were acclimated to the culture environment for at least one week before formal experiments. After the completion of the operation, the mice were executed after two spermatogenic processes, and the subsequent experiments continued on the mice.

### 2.3. Seminiferous Tubule Injection Experiment

ICR Mice were anesthetized by intraperitoneal injection (sodium pentobarbital). On the microoperation table, the reproductive organs were exposed with surgical scissors and surgical tweeds, and *KNL1* siRNA (60 µg) mixed with Entranster in vivo (Engreen Biosystem, Beijing, China) solution, added with bromophenol blue as an indicator, was injected into the seminiferous tubules of the mice. After completion of the operation, the wounds of the mice were sutured.

### 2.4. Sample Processing for HE Staining and CASA

After two spermatogenic cycles, mice were executed and testes were removed for the first recording of phenotypic data such as length, width, and surface area. The testis and epididymis were collected and placed in a fixative solution, sectioned, and stained with HE.

The epididymis was collected and cut, and the sperm was extruded by gently squeezing the epididymis with ophthalmic forceps. Then, the mixture was incubated at 37 °C, 5% CO_2_, and saturated humidity for 20 min. After the semen was completely liquefied, the mixture was evenly added to the corresponding counting board for detection for CASA detection (Hamilton Thorne, Beverly, MA, USA, TVOS) using the following parameters: VCL cutoff of 36 M/s, VSL cutoff of 5 µm/s, VAP cutoff of 20 µm, progressive minimum PR cutoff of 32%. At least 200 sperm were analyzed, and the results of the two analyses passed the statistical 95% CI. The remaining testes and epididymis tissues, which were left over after the experiment above, were frozen at −20 °C.

### 2.5. Flow Cytometry Experiment

The testis was dissected out, precooled, and washed once with 1X PBS (Solarbio, Beijing, China). We removed the tunica albuginea of the testis and fully cut up testicular tissue into the EP tube. Type II collagenase (Life-iLab, Shanghai, China) was added in 1 mL DMEM/F12 to a final concentration of 1.5 mg/mL, and the chopped testicular tissue was transferred to 37 °C and digested with 50 r/min shaking for 20 min. Then, the supernatant was removed by centrifugation at 1500 r/min for 5 min. Type II collagenase and hyaluronidase (MERK, Kenilworth, NJ, USA) were added to 0.5% trypsin to a final concentration of 1.5 mg/mL. We added 500 µL to resuspend cell mass and digested for 20–30 min at 37 °C with shaking at 150 r/min. Digestion was terminated by adding 500 µL DMEM/F12 and centrifuging at 1500 r/min for 5 min to remove the supernatant. We added 500 µL PBS (containing RNaseA (50 µg/mL)) and incubated at 37 °C for 30 min, and centrifuged at 1500 r/min for 5 min to remove the supernatant. The samples were washed once with PBS and centrifuged at 1500 r/min for 5 min to remove the supernatant.

The working mixture of Tritonx-100 + PI (biosharp, Shanghai, China) was added: PI to the final concentration of 50 µg/mL, Tritonx-100 (Sigma-Aldrich, Saint Louis, MO, USA) to the final concentration of 0.03%, and cells were drilled and stained for 10 min at room temperature in the dark. After all the treatment, the mixture was filtered with a 200 mesh filter and test on the machine.

### 2.6. TUNEL and PNA Method

The mice were killed by cervical dislocation, and the needed testicular tissue was removed and cut into a small piece 2–3 mm thick to facilitate the penetration of the fixative. The cut testicular tissue was washed with 1X PBS, fixed in the testicular tissue fixation fluid (Servicebio, Wuhan, China), and sent to the Servicebio Company (Wuhan, China) for TUNEL and PNA detection.

### 2.7. Immunofluorescence Staining

The sections or cell slides were washed with 1X PBS. They were fixed with 4% paraformaldehyde (Solarbio, Beijing, China) for 30 min at room temperature and washed three times with 1X PBS for 3 min each time. Membranes were permeabilized with Triton X-100 for 30 min at room temperature and washed three times with PBS for 3 min each. The cells were blocked overnight at 4 °C in 3% BSA (Solarbio, Beijing, China). The primary antibody was diluted, incubated at 37 °C for 1 h, and washed three times with 1X PBS for 5 min each time. The secondary antibody was diluted, incubated at 37 °C for 1 h, and washed three times with PBS for 5 min each time. Nuclear staining was performed with DAPI (Beyotime, Shanghai, China) and observed with an Olympus FV 3000.

### 2.8. siRNA Transfection and Synchronization

The cells were seeded into six-well plates a day earlier to maintain a cell density of 60%–80%. Then, 125 μL DMEM medium without antibiotics and serum was added to the EP tubes, followed by 100 pmol siRNA, blowing and mixing, then 4 μL Lipo8000™ (Beyotime, Shanghai, China) transfection reagent was added, blowing and mixing to mix well, and placed at room temperature for 20 min. Drops were added to six-well plates for cellular siRNA interference. Some cells were used for qRT-PCR to detect the efficiency of *KNL1* interference. For synchronization, after 16 h of interference in the above siRNA interference, the cells were cultured with 10 mM thymidine (Tsbiochem, Shanghai, China) for 16 h. Then, we released the thymidine to replace the medium as new and the immunofluorescence observation was performed 10 h later.

### 2.9. siRNA Sequence and Immunofluorescence Antibody

The siRNA sequence used in the cell interference experiment is Mus-si*KNL1*(5′–3′: UCGAGUCAGCUUUGCAGAUACUAUA) ordered from GENEray Biotechnology. The qRT-PCR sequence used in the experiment is *KNL1*-F: CAAAACCGAAAACTGCAGGGC; *KNL1*-R: TTTGGCTCAAGACAGCTTACC. The immunofluorescence antibody in the experiment was shown in the Appendix A.

## 3. Results

### 3.1. The Loss-Function of the KNL1 Gene Influences Male Reproductive Phenotype in Mice

We extracted the expression of the *KNL1* gene from the single-cell RNA-seq data database provided by Tong’s team, which uncovered dynamic processes in mouse spermatogenesis, revealing extensive and previously uncharacterized dynamic processes and molecular markers in gene expression [37]. Interestingly, we found that the gene *KNL1* expression is extremely high during spermatogenesis (Figure 1), implicating that the *KNL1* gene might play an important role in spermatogenesis.

We exposed the testis to injected siRNA (60 µg) in seminiferous tubules of ICR mice to influence the *KNL1* expression in the testis (Figure 2A). After two whole spermatogenic cycles, we executed mice that received no interference (*KNL1*-Ctrl mice) and *KNL1* experiment mice (*KNL1*-60 µg mice) to test different phenotype indicators. We found that in *KNL1*-60 µg mice, the size of the testis tended to decrease a little compared to *KNL1*-Ctrl mice (Figure 2B,C).

The HE staining section of the testis, testicular biopsy, and sperm between *KNL1*-Ctrl mice and *KNL1*-60 µg mice revealed changes in phenotypic characteristics. The testicular biopsy showed the defect of germ cell development in *KNL1*-60 µg mice compared to *KNL1*-Ctrl mice (Figure 3A). It was obvious that *KNL1*-Ctrl mice testis sections showed normal cell populations within the seminiferous epithelium, whereas germ cells lost most of the important developmentally relevant cells in *KNL1*-60 µg mice testis. Similarly, in the epididymal, HE staining section (Figure 3B), a marked decrease in sperm count was observed in *KNL1*-60 µg mice. Dramatically, the *KNL1*-60 µg mice had no mature spermatozoa or round spermatozoa compared with *KNL1*-Ctrl mice (Figure 3C).

### 3.2. The Loss-Function of the KNL1 Gene Results in Oligospermia and Asthenospermia in Mice

For further research, we sought to test sperm motility by CASA (computer-aided sperm analysis) detection. The mice CASA result showed the sperm count had a significant decrease in the total count, progressiveness, and motility in the *KNL1*-60 µg mice (Table 2 and Figure 4A). Although there was no significant difference in the absolute number of static sperm between the two groups (Table 3, Figure 4A,B), there was also a significant increase in percentage (relative to total sperm count), which was harmful to their breeding in the future.

All of these parameters were consistent with the diagnostic criteria for asthenospermia and oligospermia. Thus, it could be summarized the loss-function of *KNL1* leads to oligospermia and azoospermia in mice and cause harm to male reproduction.

### 3.3. Loss-Function of KNL1 Causes Spermatocytes Arrest at the Meiotic Prophase I of Spermatogenesis

The above results showed a significant decrease in sperm count and quality. We examined the integrity of sperm acrosomes by detecting peanut lectin (PNA, Figure 5A,B) and the results showed that the integrity of *KNL1*-60 µg mice sperm acrosome was significantly reduced. Next, we performed the TUNEL experiment on the section of spermatogenic tubules to prove that the degenerative phenotype of reproduction in male mice was related to apoptosis. However, to our surprise, there was no significant difference between the *KNL1*-Ctrl mice and the *KNL1*-60 µg mice (Figure 5C,D), which confirmed that apoptosis was not the main factor inducing reproductive degeneration phenotype of oligospermia and azoospermia, as many previously reported [38,39,40].

To further determine how *KNL1* blocks spermatogenesis in mice, we performed flow cytometry on their spermatozoa as Miki [30] did. The FC (flow cytometry) results revealed (Figure 6A–C) that there was normal sperm cell composition in *KNL1*-Ctrl mice and the proportion of haploid sperm cells (N) was the largest part (77.7%, *n* = 3). However, in the *KNL1*-60 µg group mice, the proportion of haploid sperm cells decreased significantly. The proportion of haploid sperm cells decreased by 49.5% (*n* = 3), the proportion of diploid sperm cells (2N) increased by 53.0% (*n* = 3), and the proportion of tetraploid sperm cells (4N) decreased by 3.6% (*n* = 3).

Considering the FC results and the spermatogenic process, we speculated that *KNL1* influenced the process of meiosis of spermatogenesis, leading to the accumulation of diploid (2N) and tetraploid (4N) sperm cells and the failure of haploid (N) sperm production.

To precisely confirm which stage the *KNL1* would affect in the spermatogenic cycle, we chose different fluorescence markers to map the cycle of *KNL1* influenced during spermatogenesis. By the detection of fluorescence intensity of different markers such as *GFRα1*, *PLZF*, *C-kit*, *STRA8*, *γH2AX*, and *Sycp3*, we found that *GFRα1*, *PLZF*, *γH2AX*, and *Sycp3* had significant differences in mouse testis between the *KNL1*-60 µg group and *KNL1*-Ctrl mice (Figure 7A–H and Appendix A).

These results again confirmed that the damage caused by *KNL1* deletion to spermatogonia mainly lies in the meiosis stage, especially affecting the transition stage of the meiosis I stage and partially affecting mitosis. We thus put forward the hypothesis that the oligospermia of *KNL1* mice mainly resulted from the loss-function of *KNL1* on spermatocyte meiosis, resulting in the block of the maturation process of spermatocytes.

### 3.4. The Loss-Function of KNL1 Leads to Abnormal Assembly and Separation of the Spindle Resulting in Unequal Cell Segregation

To further investigate the effect of *KNL1* on spermatocyte maturation, GC-2 cells that could maintain the stage of spermatocyte were selected for this study. The expression level of *KNL1* in GC-2 cells was knocked down by siRNA (Appendix A), and the GC-2 cells were treated with paving and immunofluorescence after interference. The results showed that GC-2 cells treated with *KNL1* siRNA exhibited unequal division compared to normal GC-2 cells (Figure 8).

Considering the previous reports of the *KNL1* gene in the literature, we first speculated that *KNL1* may lead to unequal division by affecting the spindle. We also transfected cells with siRNA to interfere with the expression of *KNL1*. Immunofluorescence results showed that compared with the non-interference group, the spindle in the *KNL1*-60 µg group represented different degrees of abnormality compared to the *KNL1*-Ctrl group (Appendix A), such as the emergence of multipolar spindles in the metaphase and the failure of spindle formation (Figure 9A,B and Appendix A).

In addition, we also observed the phenomenon of chromosomal abnormalities in interfered cells. Chromosomes were not fully aligned on the equatorial plate in metaphase, and there was a phenomenon of chromosome lag in anaphase (Figure 9C,D).

We inferred that the disorder of the spindle leads to the decline of its ability to regulate chromosomes and the spindle cannot regulate chromosomes as accurately as before, which resulted in the abnormal arrangement of some chromosomes based on the evidence above. The force imbalance causes chromosome lag in the subsequent cell separation and leads to abnormal cell division.

## 4. Discussion

Maintaining reproductive health is crucial to successful fertility and a healthy relationship between partners, while most cases of infertility originate from the poor production of sperm or eggs [41]. Nowadays, male infertility is a big problem plaguing mankind, which may be caused by genetic factors or an unhealthy lifestyle [42]. There are several factors such as genetic mutations, infections, anatomical change, and pressure involved in the normal development and quality of sperm that stop men from being able to father offspring [43]. The phenotype of nearly 90% of male infertility is mainly reflected in sperm, including sperm quantity, quality, and activity [44].

*KNL1* is an essential large scaffold protein for accurate chromosome segregation in eukaryotic cells [16]. It was first recognized as a kinetochore protein in 2004 in human cells [45]. Subsequently, more and more studies have gradually revealed the functions and corresponding connections of different domains and motifs of *KNL1* [9,12,46,47,48]. Previous studies have shown that *KNL1* plays important roles in several types of cancer, and *KNL1* was found to be abnormally expressed in 16 types of cancer [49,50]. However, its relationship with the male reproduction system remains largely unclear.

In our study, the expression of *KNL1* in the testis was knocked down by injecting the corresponding siRNA into the seminiferous tubules of the mice to explore the influence on the male reproductive phenotype. To our surprise, no obvious changes in testicular size were observed in NC and *KNL1* groups. However, when we detected the internal physiological structure of the testis and epididymis, we found the *KNL1* group lost most of the important developmentally relevant cells and there is a significant decrease in the number of sperm (Table 2 and Table 3 and Figure 4A–D). Compared to the control group, the quality and quantity of sperm both have a huge reduction. These phenotypes are consistent with those previous studies concerning oligospermia and asthenospermia due to gene variants, oxidative stress, or stimulation of sexual hormone secretion [51,52,53,54]. In the flow cytometry, we found that the number of diploid cells in the experimental group was significantly accumulated, suggesting that there was a block in the process of spermatogenesis, leading the number of mature sperm to decrease. At this point, these phenotypes were already consistent with the diagnosis of oligospermia and asthenospermia.

Next, the effect and underlying mechanism of this block in spermatogenesis were investigated. Previous researchers found that apoptosis might be one of the important reasons leading to sperm problems [38,39,40]. However, it was surprising to find that apoptosis was not the main factor causing sperm reduction and sperm arrest. Immunofluorescence experiments were performed to detect the markers in each stage of spermatogenesis to precisely locate the problem period and we found that the attenuation of the fluorescence signal was most obvious in the meiotic phase, which combined with the previous flow cytometry results. We believed that the problem of sperm arises in the first meiotic division. GC-2 cell as a cell model was adopted to explore the specific effect and underlying mechanism after *KNL1* interference. We found that unequal division of GC-2 cells occurred after the knockdown of *KNL1* expression in GC-2 cells. Since previous reports have confirmed the role of *KNL1* in the regulation of kinetosomes during cell division, we hypothesized that *KNL1* also has a certain effect on the spindle. Consistent with our hypothesis, there was a phenomenon of multipolar spindle and spindle dispersion after observing *KNL1* knockdown cells by immunofluorescence. Additionally, chromosome disarrangement and chromosome hysteresis were also observed. Taken together, the knockdown of *KNL1* could lead to decreased adhesion to the spindle, resulting in the abnormal spindle and decreased tensile force on the chromosome, abnormal division in cells, and increased probability of aneuploidy in cells. For male reproduction, it can lead to abnormal spermatogenesis, oligospermia, and asthenospermia.

To conclude, we first revealed the association between the *KNL1* gene and spermatogenic dysfunction and further explored the mechanism of *KNL1* in causing the occurrence of spermatogenic dysfunction. *KNL1* can become a potential molecular prognostic and diagnostic marker in some cases while normal phenotypes are easily ignored. What is more, our work not only further reveals the role of the *KNL1* gene, but also provides a more accurate tool for detecting the abnormal stages of spermatogenesis by using FCM and IF, as well as expanding the diagnostic pointer pool in the field of male reproduction. As a result, it can improve diagnostic accuracy if coupled with other clinical diagnostic technologies. However, some limitations should be noted. Firstly, although we identified the problem stage in the meiosis I stage, we did not pinpoint the different stages in meiosis such as the leptotene stage, or even the zygotene stage, pachytene stage, diplotene stage, and diakinesis phase. Other innovative technology to further explore more accurate positioning is worth further exploration in our experiment that we should continue to explore. Secondly, establishing an animal model of the cKO mice to explore the mechanism of *KNL1* in more detail in subsequent studies should be considered. In addition, to date there has been no reported human case associated with *KNL1*, and future studies in the reproductive field should be paid attention to establish a strong link between *KNL1* and human sperm quality and health.

## Figures and Tables

**Figure 1 sensors-23-02571-f001:**
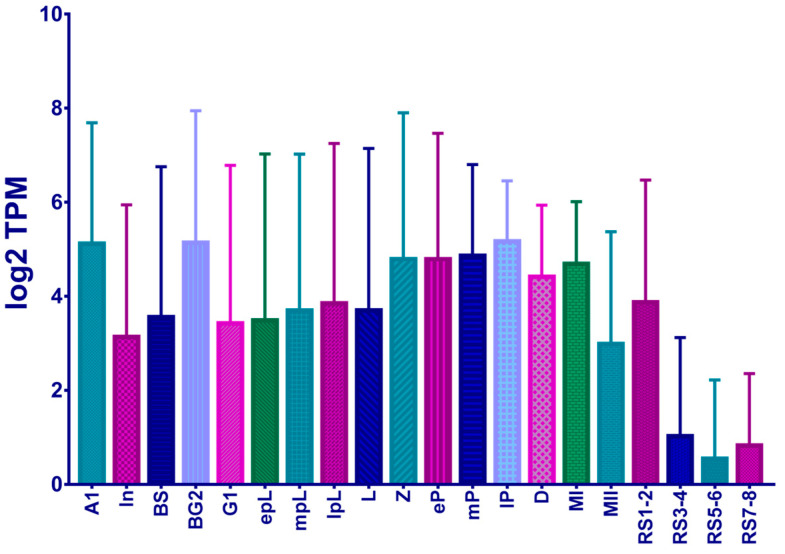
Log2 TPM values in different spermatogenic stages. A1 represents the type A1 spermatogonia. In represents the Intermediate spermatogonia. BS represents the Type B spermatogonia S phase. BG2 represents the Type B spermatogonia G2M phase. G1 represents the G1 phase of the preleptotene spermatocyte stage. ePL represents the early phase of the preleptotene spermatocyte stage. mPL represents the medium-term phase of the preleptotene spermatocyte stage. IPL represents the late phase of the preleptotene spermatocyte stage. L represents the leptotene stage of meiosis. Z represents the even line stage of meiosis. eP represents the early pachytene of meiosis. mP represents the metaphase pachytene of meiosis. IP represents the late pachytene of meiosis. D represents the double-line stage of meiosis. MI represents the first meiotic division. MII represents the second meiotic division. RS1–2 represents the round spermatid stage 1–2. RS3–4 represents the round spermatid stage 3–4. RS5–6 represents the round spermatid stage 5–6. RS7–8 represents the round spermatid stage 7–8.

**Figure 2 sensors-23-02571-f002:**
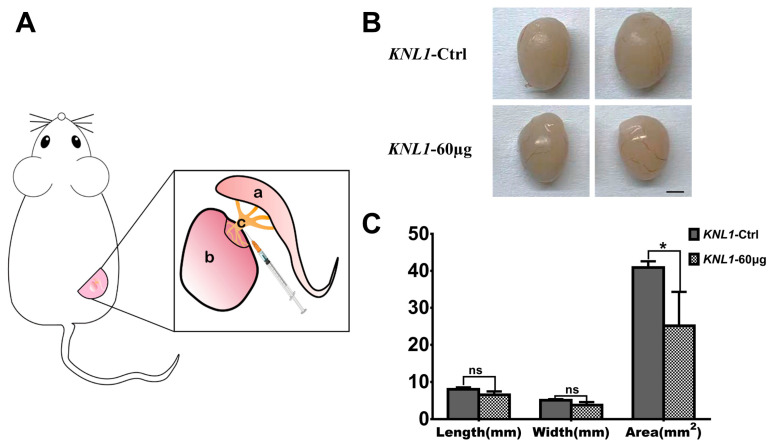
Operation diagram and the loss-function of *KNL1* to the testicle phenotype. (**A**) Surgical model of injecting spermatogenic tubules in mice. a is the epididymis; b is the testis; c is the injection site in spermatogenic tubules. (**B**) The testicular phenotype of *KNL1*-Ctrl mice and *KNL1*-60 µg mice. bar = 1 mm. (**C**) The degree of difference between the *KNL1*-Ctrl mice and *KNL1*-60 µg mice. Data were presented as mean percentages (mean ± SEM) of at least three independent measurements. Asterisk denotes statistical difference level of significance (*, *p* < 0.05; ns > 0.05).

**Figure 3 sensors-23-02571-f003:**
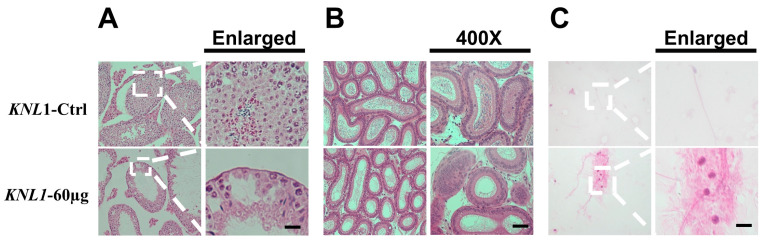
HE staining sections of loss-function of *KNL1* on the phenotype of male mice. (**A**) The HE staining of testis sections between *KNL1*-Ctrl mice and *KNL1*-60 µg mice. The right represents the magnified portion of the white dashed box in the left line. Bar = 20 µm (**B**) The HE staining of epididymis sections between *KNL1*-Ctrl mice and *KNL1*-60 µg mice. The left is a 200× magnification and the right is a 400× magnification bar = 50 µm. (**C**) HE staining section of sperm between *KNL1*-Ctrl and *KNL1*-60 µg mice. The right represents the magnified portion of the white dashed box in the left line. Bar = 2 µm.

**Figure 4 sensors-23-02571-f004:**
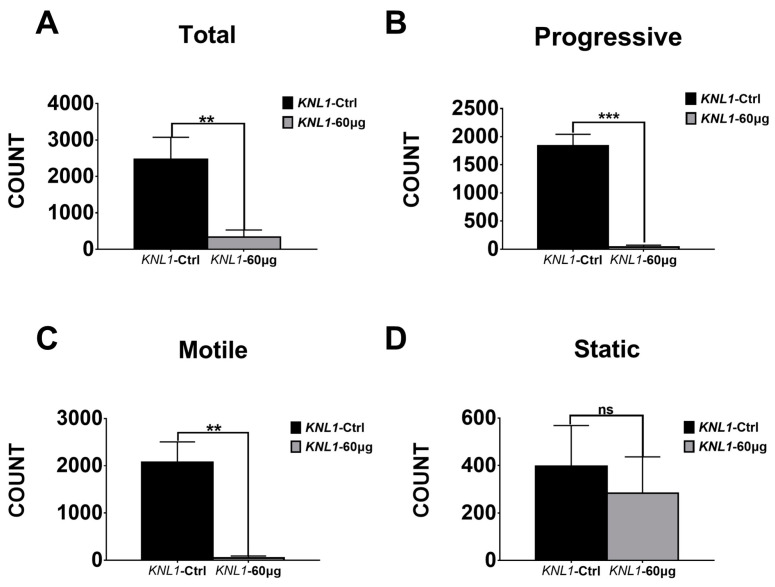
The CASA data of *KNL1*-Ctrl and *KNL1*-60 µg Group mice. (**A**) CASA total detection of *KNL1*-Ctrl and *KNL1*-60 µg. (**B**) CASA progressive detection of *KNL1*-Ctrl and *KNL1*-60 µg. (**C**) CASA motile detection of *KNL1*-Ctrl and *KNL1*-60 µg. (**D**) CASA static detection of *KNL1*-Ctrl and *KNL1*-60 µg. Asterisk denotes statistical difference level of significance (**, *p* < 0.01; ***, *p* < 0.001, ns > 0.05).

**Figure 5 sensors-23-02571-f005:**
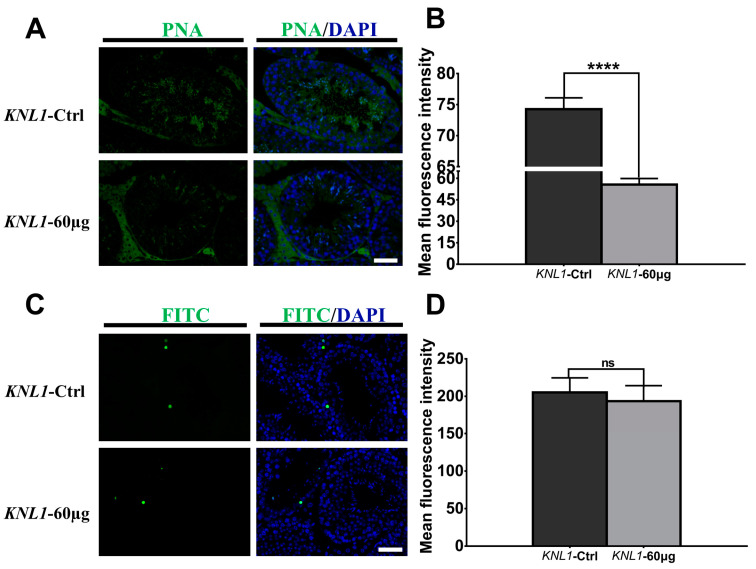
Effects of loss-function of *KNL1* on acrosome and apoptosis. (**A**) The PNA immunofluorescence of a section of the testicle. Testis was immunostained with anti-PNA (green). Merge was DAPI (blue) and PNA (green). The top is *KNL1*-Ctrl mice and the bottom is *KNL1*-60 µg mice. Bar = 50 µm. (**B**) Mean immunofluorescence intensity of PNA between *KNL1*-Ctrl mice and *KNL1*-60 µg mice. (**C**) The TUNEL assay of a section of the testicle. The FITC immunofluorescence of a section of the testicle. Testis was immunostained with anti-FITC (green). Merge was DAPI (blue) and FITC (green). The top is *KNL1*-Ctrl mice and the bottom is *KNL1*-60 µg mice. Bar = 50 µm. (**D**) Mean immunofluorescence intensity of TUNEL between *KNL1*-Ctrl mice and *KNL1*-60 µg mice. All data were at least measured by three independent experiments. Asterisk denotes statistical difference level of significance (****, *p* < 0.0001, ns > 0.05).

**Figure 6 sensors-23-02571-f006:**
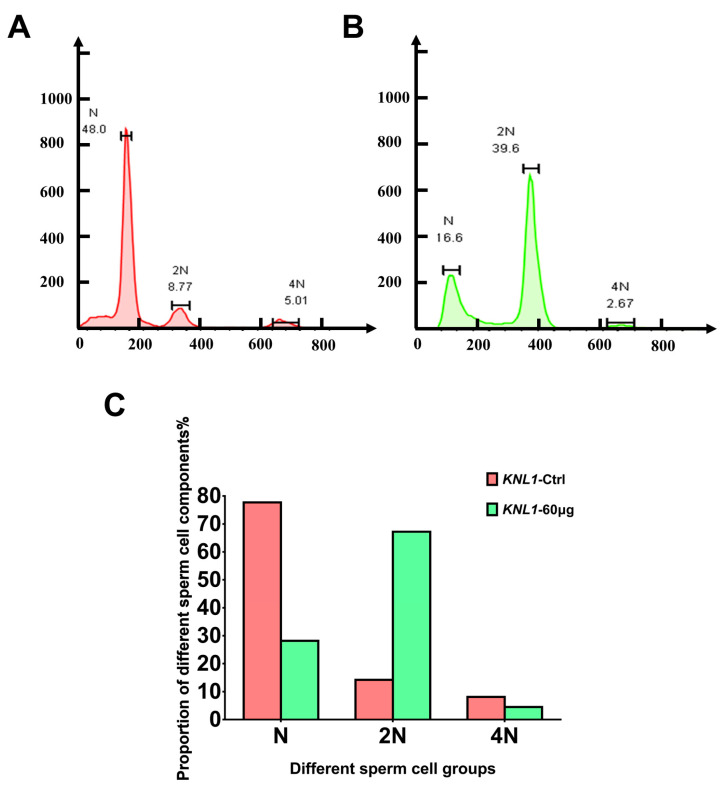
Effects of loss-function of *KNL1* on the spermatogenic process by FCM. (**A**) The FCM results of *KNL1*-Ctrl mice. (**B**) The FCM results of *KNL1*-60 µg mice. N, 2N, and 4N represents a different period of sperm in spermatogenesis (**A**,**B**). (**C**) The percentage of sperm with different ploidy. All data were at least measured by three independent experiments.

**Figure 7 sensors-23-02571-f007:**
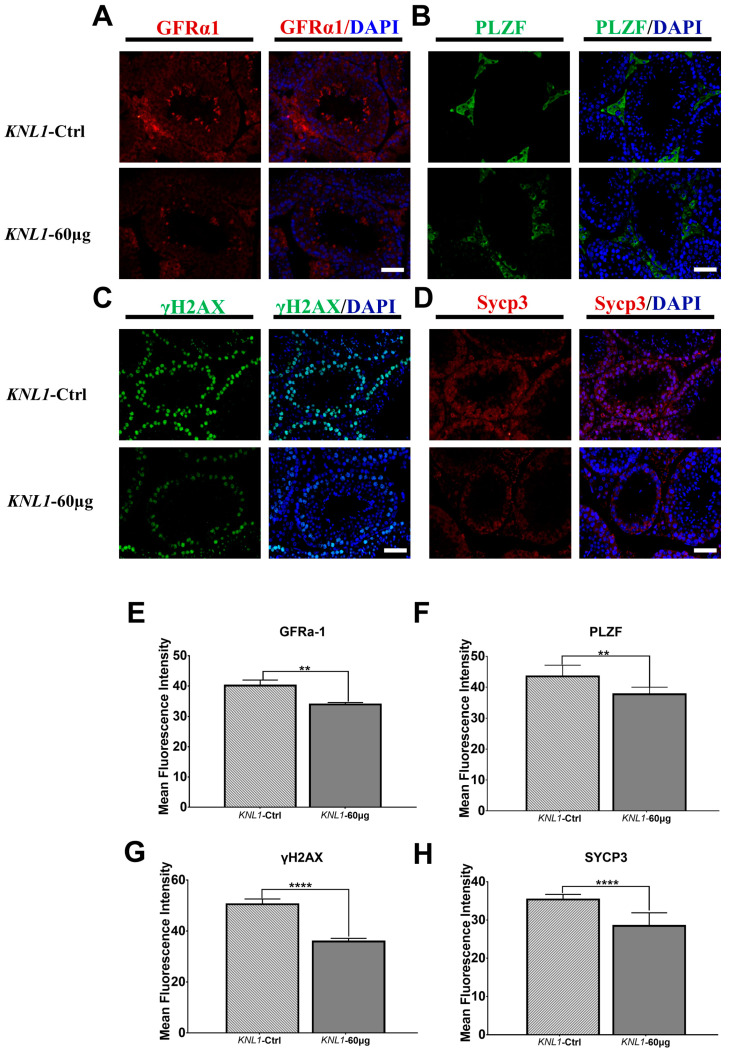
Immunofluorescence of four markers between *KNL1*-Ctrl and *KNL1*-60 µg. (**A**) GFRα1 fluorescent images of testis in *KNL1*-Ctrl and *KNL1*-60 µg. Testis was immunostained with anti-GFRα1 (red). Merge was DAPI (blue) and GFRα1 (red). (**B**) PLZF fluorescent images of testis in *KNL1*-Ctrl and *KNL1*-60 µg. Testis was immunostained with anti-PLZF (green). Merge was DAPI (blue) and PLZF (green). (**C**) γH2AX fluorescent images of testis in *KNL1*-Ctrl and *KNL1*-60 µg. Testis was immunostained with anti-γH2AX (green). Merge was DAPI (blue) and γH2AX (green). (**D**) SYCP3 fluorescent images of testis in *KNL1*-Ctrl and *KNL1*-60 µg. Testis was immunostained with anti-SYCP3 (red). Merge was DAPI (blue) and SYCP3 (red). (**E**–**H**) Data were presented as mean percentages (mean ± SEM) of at least three independent measurements. Asterisk denotes statistical difference at a *p* (**) < 0.0 1, *p* (****) < 0.0001 level of significance. Bar = 50 µm.

**Figure 8 sensors-23-02571-f008:**
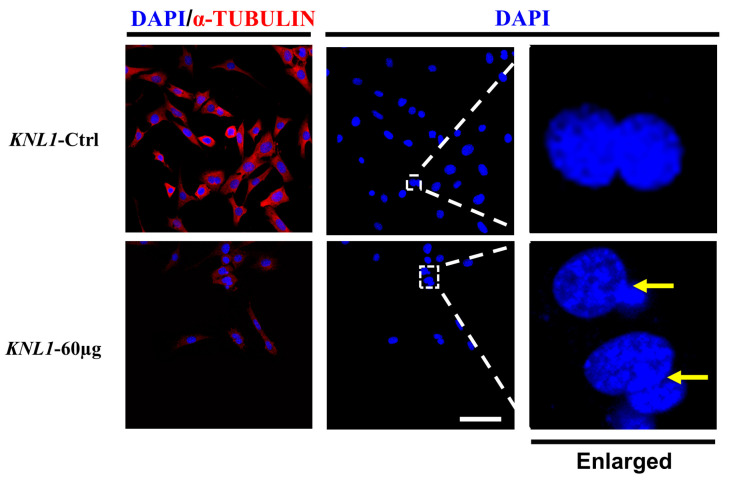
Effect of depletion of *KNL1* on GC-2 cell separation. Fluorescent images of GC-2 cells interfered with by *KNL1* siRNA. GC-2 cells were immunostained with anti-α-tubulin (red) and DAPI (blue). Merge was DAPI (blue) and anti-α-tubulin (red). The right represents the magnified portion of the white dashed box in the middle. The yellow arrow shows the cells that divide unequally. Bar = 50 µm.

**Figure 9 sensors-23-02571-f009:**
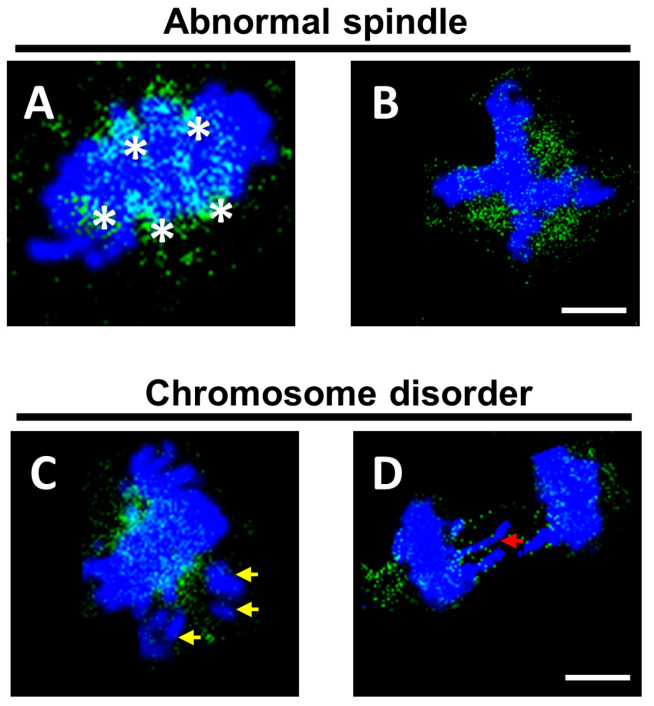
Effect of depletion of *KNL1* on GC-2 cell spindle assembly and chromosome. (**A**,**B**) The abnormal spindle after interfering with *KNL1*. Asterisks represent different spindle poles. (**C**,**D**) The abnormal chromosome arrangements after interfering with *KNL1*. Yellow arrows represent chromosomes not aligned on the equatorial plate, red arrows represent lagged chromosomes. Green is α-tubulin, and blue is DAPI. Bar = 5 µm.

**Table 1 sensors-23-02571-t001:** Part of the literature on the changes of FCM used in sperm detection.

Researchers	Year	Methods	Machine Type	Detection
Miki Hara-Yokoyama et al. [30]	2019	FCM + PI	FACSCalibur	The proportion of haploid, diploid, and tetraploid cells during spermatogenesis
Xianrong Xiong et al. [32]	2021	FCM + Hoechst33342 + Allura Red	MoFlo XDP	Sperm X and Y were sorted out
Marc et al. [33]	2022	FCM + Simultaneous co-staining (CMA3 and Yo-Pro-1)	CytoFLEX cytometer	Sperm quality parameters (morphology, viability, total and progressive motility)
Raul et al. [34]	2021	The co-staining LD + AO	CytoFLEX S	Sperm membrane integrity and DNA fragmentation
Phillip et al. [35]	2022	FCM + a variety of fluorescent dye (PTYR, PDK, FITC-PNA, PI, M540 and Yo-Pro-1)	CYAN-ADP	Sperm capacitation and functions
Evelyn et al. [36]	2023	FCM + co-staining SytoxGreen™ and dihydroethidium (DHE)	Amnis^®^ ImageStream^®^	Certain morphologic abnormalities and ROS

**Table 2 sensors-23-02571-t002:** The CASA count number data of *KNL1*-Ctrl and *KNL1*-60 µg Group mice.

Count	*KNL1*-Ctrl Group			*KNL1* Group		
Total	3164	2071	2180	109	419	471
Motile	2569	1776	1878	0	74	75
Progressive	2067	1676	1771	0	62	54
Rapid	2496	1768	1875	0	72	75
Slow	73	8	3	0	2	0
Static	595	295	302	109	345	396

**Table 3 sensors-23-02571-t003:** The CASA percentage data of *KNL1*-Ctrl and *KNL1*-60 µg Group mice.

Percentage	*KNL1*-Ctrl Group (%)			*KNL1* Group (%)		
Motile	81.2	85.8	86.1	0	17.7	15.9
Progressive	65.3	80.9	81.2	0	14.8	11.4
Rapid	78.9	85.4	86	0	17.2	15.9
Slow	2.3	0.3	0.1	0	0.4	0
Static	18.8	14.2	13.9	100	82.3	84.1

## Data Availability

The data presented in this study are available on request from the corresponding author. The data are not publicly available due to some experimental research still in progress.

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
