# Peer review of "The Loss-Function of KNL1 Causes Oligospermia and Asthenospermia in Mice by Affecting the Assembly and Separation of the Spindle through Flow Cytometry and Immunofluorescence"

_sensors, 2023, doi:10.3390/s23052571_

Round 1

Reviewer 1 Report

In this manuscript, first linked KNL1 to male reproductive health and showed that the loss-function of KNL1 resulted in oligospermia and asthenospermia in mice through CASA. Using numerical analysis,  49.5% haploid sperm was reduced and 53.2% diploid sperm increased after the function of KNL1 was lost and identified spermatocytes arrest at the meiotic prophase I of spermatogenesis, which was induced by abnormal assembly and separation of the spindle. Although the authors extensively investigated the properties of the proposed platform, the work suffers from drawbacks that must be addressed in the revised version of the manuscript. I suggest a minor revision before the decision whether it can be accepted or not.

Additional comments:

1. The English is not good in this manuscript, there are some English problems.

2.This paper lacks the comparison of similar methods and structural performance. This can be performed by preparing a table and summarizing those details.

3. The format of formulas, numbers, variables and References in the paper needs to be unified and accurate.

4.  The introduction section is prepared in a old-fashioned and boring style.  It does not refer to the recent and innovative researches that have been reported in the literature. The Introduction must provide highlights from the fundamentals and state-of-the-art finding of the context.

Reviewer 2 Report

The article “The loss-function of KNL1 causes oligospermia and asthenospermia in mice by affecting the assembly and separation of  the spindle through flow cytometry and immunofluorescence” to investigate the disruption of the KNL1 gene in the mouse testis and the GC-2 germ cell line changes were observed in the size of the testis, its histology, decreased sperm motility, alteration of the acrosome, increase in cells in the diploid state, decrease in post-meiotic proteins, irregular cell division and alteration of the spindle. All this in the context of male fertility, which has not previously been studied for this KNL1 gene. 

Minor concerns:

1) Line 102, describes whether the solution with siRNA contained some transfection agent. 

2) Line 109 to correct Some tests  ……. Some testes 

other tests ………….other testes

3) It is suggest that KNL1 immunodetection during spermatogenesis in both control and siRNAKNL1 animals.

 4) Sperm were obtained from epididymis for testicles or CASA?  Is necessary to clear this  in 2.4 Sample processing

5) I suggest to mention in the tables the total sperm count in the control and KNL1 silenced mice to support the state in discussion:

 Lines 359 and 360 we found the KNL1 group lost most of  the important developmentally relevant cells and there is a significant decrease in the  number of sperm..

6)  It would have been more valuable to track the KNL1 protein in the micrographs to effectively know if the cells with changes in cell division are depleted of KNL1, as well as in the sections of testes treated with the KNL1 siRNA. And it might also have been helpful to track a kinetochore protein like CENP-C, T or Ndc80C during the experiments.

Reviewer 3 Report

The manuscript strives to study the role of KNL1 in the loss of sperm production and/or function. The assessment of different proteins present in the structures critical for sperm survival and activity is nowadays a highly up-to-date subject and a choice to localize KNL1 within the male reproductive tract and cells and to study its involvement in the maintenance of spermatogenesis and sperm motility presents a modern approach.

The manuscript is well structured and presents with a comprehensive approach to evaluate the study hypothesis. Several advanced methods has been applied to reach the goals of the study, providing a complex analysis. The introduction part provides sufficient background. The aim is clear, and methodology is appropriate. The presentation of the results is comprehensive, supported by multiple figures and supplementary files. The discussion is concise and to the point, highlighting the most important aspects of this study.

I have only several minor remarks concerning the form of the manuscript and besides them, I find the manuscript valuable to be published in the scientific journal Sensors.

Remarks:

1. What does “KNL1” stand for? Please explain the abbreviation.

2. What was the origin of GC-2 cells? Were these purchased or collected elsewhere?

3.  Line: What do “other tests” mean?

4. Please, magnify all figures in the main body since these are very difficult to read. At the same time, what does “A” in Figure 1 mean?

5. What CASA instrument was used for the evaluation of the sperm motility? What were the cut off values for the evaluation?

6. The authors could briefly discuss any limitations of this study.
